# Current provision of myelopathy education in medical schools in the UK: protocol for a national medical student survey

Oliver Mowforth [1], Benjamin Davies,[1] Max Stewart,[2] Sam Smith,[2] Alice Willison,[3] Shahzaib Ahmed,[2] Michelle Starkey,[4] Iwan Sadler,[4] Ellen Sarewitz,[4] Sybil Stacpoole,[5] Mark Kotter[1,6]

World Federation of Neurosurgical Societies International Meeting, Belgrade, Serbia, March 2019. British Neurosurgical Research Group Meeting, Edinburgh, November 2019.

For numbered affiliations see end of article.

**Correspondence to**
Dr Mark Kotter;
mrk25@cam.ac.uk

## ABSTRACT

**Introduction** Degenerative cervical myelopathy (DCM) is a common, disabling and progressive neurological condition triggered by chronic compression of the cervical spinal cord by surrounding degenerative changes. Early diagnosis and specialist management are essential to reduce disability, yet time to diagnosis is typically prolonged. Lack of sufficient representation of DCM in undergraduate and postgraduate medical curricula may contribute to the poor recognition of DCM by non-specialist doctors in clinical practice.

In this study, our objective, therefore, is to assess DCM teaching provision in medical schools throughout the UK and to assess the impact of teaching on the DCM knowledge of UK medical students.

**Methods and analysis** A 19-item questionnaire capturing data on medical student demographics, myelopathy teaching and myelopathy knowledge was designed. Ethical approval was granted by the Psychology Research Ethics Committee, University of Cambridge. An online survey was hosted on Myelopathy.org, an international myelopathy charity. Students studying at a UK medical school are eligible for inclusion. The survey is advertised nationally through university social media pages, university email bulletins and the national student network of Myelopathy.org. Advertisements are scheduled monthly over a 12-month recruitment period. Participation is incentivised by entering consenting participants of completed surveys to an Amazon voucher prize draw. Responses are anonymised using participant-chosen unique identifier codes. A participant information sheet followed by an explicit survey question captures participant informed consent. Regular updates on the progress of the study will be published on Myelopathy.org.

**Ethics and Dissemination** Ethical approval for the study was granted by the Psychology Research Ethics Committee, University of Cambridge (PRE.2018.099). The findings of the study described in this protocol, and all other related work, will be submitted for publication in a peer-reviewed journal and will be presented at scientific conferences.

### Strengths and limitations of this study

► This is a large national questionnaire-based study targeted at all medical students in the UK.
► This is the first study to assess degenerative cervical myelopathy (DCM) medical education in the UK.
► This is a student-led study with oversight from senior researchers and involvement of patients with DCM.
► The study data will provide insight into an area that has potential to translate into substantial improvements in patient outcomes.
► Risk of selection bias may overestimate the current knowledge of DCM among UK medical students.

symptomatic cervical spinal cord compression due to degenerative changes of the spine (figure 1).[1] DCM is the most common cause of spinal cord impairment in adults worldwide, with an estimated prevalence of up to 5% in the over 40s.[1–3]

Substantial improvements in the care of patients with DCM could be realised today without any additional scientific understanding of the condition.[1] Patients with DCM almost universally experience diagnostic delays. For example, one study demonstrated that on average it took 2 years and five clinic appointments before the diagnosis was made.[4] Improvement in diagnosis is therefore a key target that would provide immediate reductions in disability.[5]

Initially, patients with DCM are commonly seen by primary care, emergency and medical physicians.[6 7] These non-specialist triage points appear key to earlier diagnosis, yet DCM features poorly in medical curricula.[8] This risks non-specialist doctors being poorly equipped to detect DCM.

The current focus and makeup of DCM medical education are unclear. Our recent

## INTRODUCTION

Degenerative cervical myelopathy (DCM) is a common neurological condition of

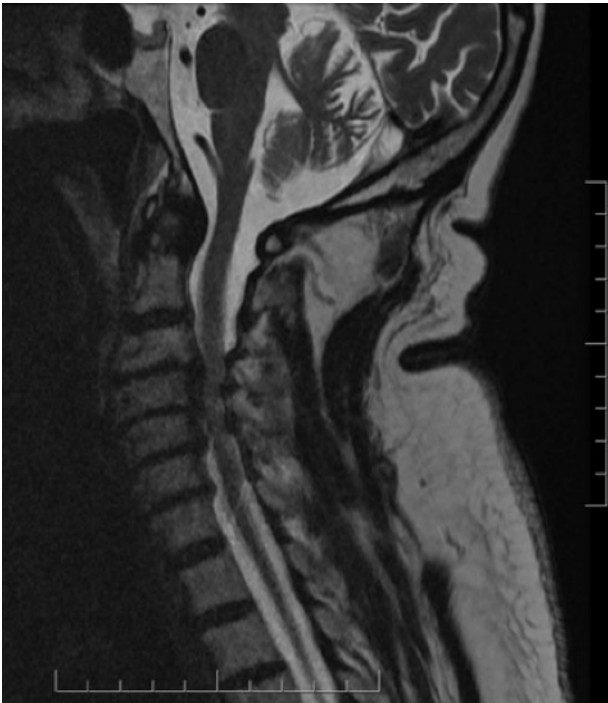

**Figure 1** Typical MRI scan from patient with degenerative cervical myelopathy showing spinal cord compression and multilevel degenerative changes.

evaluation of undergraduate and postgraduate medical curricula identified deficiencies in DCM representation, however, collectively this did not correlate with student performance in multiple-choice question banks.[8]

The aim of this study is to describe the teaching of DCM in UK medical schools and to assess the influence on medical student knowledge of DCM. Ultimately, the aim is to identify both training and knowledge gaps to inform guidance on DCM education in order to address the burden and impact of delayed diagnosis. We hypothesise that knowledge and awareness among medical professionals are related to their specific medical education on DCM.

## METHODS AND ANALYSIS
### Study partners
#### Myelopathy.org
Myelopathy.org is an international cervical myelopathy charity, which aims to improve health and wellbeing in DCM.[9] This study forms part of a wider project to evaluate DCM education and awareness among medical professionals in the UK and develop focused and effective interventions to improve this. This includes our recent gap analysis of key medical resources and knowledge at each stage of the primary care training pathway[8] and an educational article on DCM for non-specialist medical professionals, commissioned by the *British Medical Journal* (figure 2).[1] The charity is focused on understanding how medical education can be improved as one possible strategy to address the current problems of delayed, missed and underdiagnosis.[1]

#### Student Society of Myelopathy.org
The foundations of professional knowledge begin with students' formative experiences at medical school. Therefore, the Student Society of Myelopathy.org was established in 2018 with the aim of improving DCM engagement and awareness among medical students. The society remains in its infancy but has hosted a series of educational talks at the University of Cambridge; established national medical student essay and research prizes; designed and

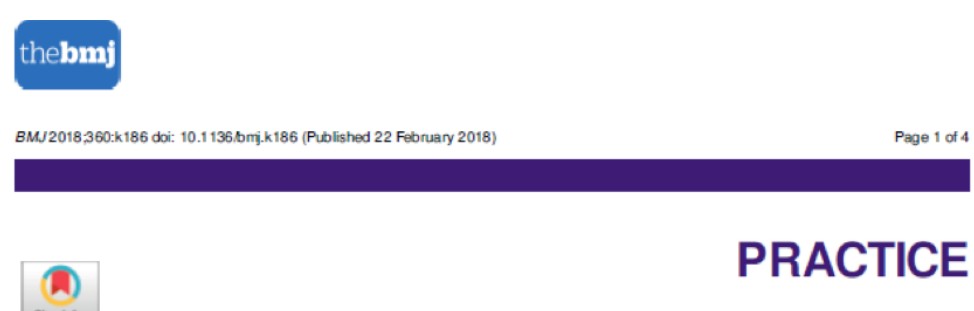

BMJ 2018;360:k186 doi: 10.1136/bmj.k186 (Published 22 February 2018)          Page 1 of 4

**PRACTICE**

EASILY MISSED?

## Degenerative cervical myelopathy

Benjamin M Davies *academic clinical fellow and specialist registrar neurosurgery*[1], Oliver D Mowforth *medical student,*[1], Emma K Smith *specialist registrar general practice,*[2], Mark RN Kotter *clinician scientist*[1] *and consultant neurosurgeon*[1]

[1]Academic neurosurgery unit, Department of Clinical Neurosurgery, University of Cambridge, Cambridge, UK; [2]School of General Practice, NHS Health Education East of England, UK

**Figure 2** Degenerative cervical myelopathy educational article commissioned by the *British Medical Journal*. Reproduced with permission from *British Medical Journal*.

initiated this current study and secured sponsorship from corporate partners to support these projects.

## Study design

An electronic survey was agreed to be the most efficient way to simultaneously assess teaching, perceptions of teaching and current knowledge and to efficiently reach students from all UK medical schools. Part 1 of the survey was designed to capture data on basic demographic characteristics and teaching on DCM in UK medical schools. Part 2 was designed to capture data on knowledge and awareness of fundamental DCM facts including prevalence, symptoms, diagnostic imaging modalities and surgical effectiveness in addition to appreciation of important current challenges such as prolonged time to diagnosis and poor patient quality of life. Part 2 was designed to capture participant's perception of their DCM knowledge. These questions were considered to be a reasonable balance of capturing sufficient data while maintaining an acceptably short survey completion time of approximately 5 min.

The study management team comprised representation from second year to final year students from multiple UK medical schools. The team comprised an academic neurosurgical registrar (BD) and an academic neurosurgical consultant (MK), both with special clinical and academic interests in DCM, and an academic consultant neurologist (S Stacpoole) who is a neuroscience teaching fellow and neuroscience course organiser at the University of Cambridge. Importantly, this is a student-motivated, student-led project with supervision, oversight and mentorship at all stages of the study from a team of senior researchers.

The survey was iteratively refined among the study management group until section 1 was felt to capture sufficient information on current DCM teaching provision in UK medical schools. Similarly, section 2 was refined until it was felt to evaluate fundamental DCM facts at a level sufficient for a non-specialist but qualified doctor to appropriately suspect, investigate, diagnose and redirect care in DCM. In total, a 19-item questionnaire was developed (figure 3).

Survey development closely aligns with the Association for Medical Education in Europe (AMEE) seven-step guide to medical education survey development: (1) literature review, (2) focus group establishment, (3) synthesis of literature review/focus group, (4) item development, (5) expert validation, (6) feedback and expert validation and (7) piloting,[10] with the exception that a quantitative gap analysis was performed in the place of a literature review due to the lack of research on DCM medical education at the time of study design.[8] Nonetheless, planning for a systematic review of similar medical education surveys in other diseases is underway.

## Survey piloting

The survey was piloted in January 2019 by a group of 20 medical students at the University of Cambridge. These students were not directly involved in the design of the study. Data were inspected to identify any inconsistencies, ambiguities and problematic survey items. No issues were identified. Students in the pilot group were contacted to ask whether they had encountered any difficulties, areas of uncertainty or suggestions for improving the survey. Students reported the survey to be acceptable on both desktop and mobile devices. Informal cognitive interviews of five students revealed that survey items were interpreted by students as intended by the study team, without ambiguity. No further alterations were therefore made. No formal statistical data analysis was conducted at the pilot stage.

## Survey administration

To maximise reach across the UK, the survey was hosted on Qualtrics survey software (Provo, Utah, USA), a professional online survey platform. Qualtrics survey software includes both desktop and mobile-compatible versions, maximising convenience for participants.

The survey comprises two pages. The first page acts as a frontpage consisting of a participant information sheet and a question capturing informed consent. The second page is the main survey page. It comprises 19 questions over a single page to minimise the number of clicks required for participants to complete the survey.

## Survey dissemination

A network of Myelopathy.org student representatives was recruited, with representation of students from 25 different medical schools across the UK. Some but not all were also representatives for the Society of British Neurological Surgeons and Neurology and Neurosurgery Student Interest Group (NANSIG). A standard advertisement devised by the protocol development team and approved by the research ethics committee is used for all advertising (figure 4). Representatives from all medical schools are sent monthly reminder emails to disseminate adverts at their institution over a 12-month period from February 2019 to January 2020. There was no difference in the timing of reminder emails between different medical schools. Although prompted to advertise monthly, the final decision on frequency of advertising at each institution is left to the discretion of representatives, guided by local rules.

On completion of the survey participants are invited to leave an email address in a separate Google Forms survey to be entered into a prize draw to win one of five £10 Amazon vouchers. The prize draw was agreed by the study management group and approved by the research ethics committee. The rationale was that prizes of this value may encourage participation without providing an inappropriate incentive to participate.

The survey is advertised in social media groups and on social media pages for relevant student groups and societies at universities across the UK. For example, Facebook groups of surgical, neurosurgical, neurological and other medical student societies were approached. In addition, university mailing lists and email bulletins are being

MYELOPATHY.ORG
STUDENT SOCIETY

**Myelopathy Student Survey**

**First Question. Consent:** I have read the participant information sheet and consent to my survey response being used in this study (yes/no).

**Survey Part 1: Teaching**

- Which medical school do you study at?
  (dropdown selection of all UK medical schools, including other category)
- Preclinical or clinical?
  (dropdown selection of preclinical or clinical)
- Which year of medical school?
  (dropdown selection: year1; year2; year3; year4; year5; year6; other)
- What is your planned/most likely career speciality?
  (free entry)
- Have you heard of degenerative cervical myelopathy (also known as cervical spondylotic myelopathy)?
  (dropdown selection: yes; no)
- Have you had any teaching on cervical myelopathy from your medical school?
  (dropdown selection: yes; no)
- What is the total duration of your cervical myelopathy teaching?
  (dropdown selection: 0 mins; 15 mins; 30 mins; 1 hour; 2 hours; >2 hours)
- What form did the teaching take?
  (dropdown selection: lecture; seminar; ward-based teaching; e-learning; other; N/A)
- How much private study have you done on cervical myelopathy?
  (dropdown selection: 0 mins; 15 minutes; 30 minutes; 1 hour; 2 hours; >2 hours)

**Survey Part 2: Knowledge**

**For many of the following questions there is often not necessarily a right or wrong and we are often more interested in your perception.**

- How would you rate your overall knowledge of cervical myelopathy?
  (dropdown selection: excellent, very good, average, poor, very poor)
- What do you estimate is the prevalence of cervical myelopathy is in the over 40s?
  (dropdown selection: 0.01%; 0.1%; 1%; 5%; 10%; 50%)
- Which do you think is the best imaging modality to diagnose myelopathy? (4 or 5 choices)
  (dropdown selection: MRI; CT; X-ray; myelography; ultrasound)
- What do you think the average time to diagnosis is from first presentation? (5 choices)
  (dropdown selection: 1 week; 1 month; 1 year; 2 years; 5 years)
- On average, how many consultations do you think there are before a patient is diagnosed?
  (dropdown selection 1;2;5;10;20)
- Do you think time to diagnosis is likely important for long-term prognosis?
  (dropdown selection: yes; no)
- What effect do you think surgery is likely to have on a patient's symptoms?
  (dropdown selection: improve; stabilise; worsen)
- Which of these symptoms might be present in cervical myelopathy?
  (clumsy gait; neck pain/stiffness; falls; imbalance; loss of bowel control; loss of urinary control; erectile dysfunction; paraesthesia; limb weakness; limb numbness; limb pain; loss of dexterity)
- Please rank these conditions from worst to best in terms of your estimate of patient quality of life?
  (dropdown selection: cancer; cervical myelopathy; diabetes; depression; heart failure)
- How would you most like to learn about cervical myelopathy in the future?
  (dropdown selection: lecture; seminar; private study; online e-learning material; not interested)

**Figure 3** Final survey design. Reproduced with permission from Myelopathy.org.

used, where appropriate, in accordance with local rules. Adverts are also placed in the monthly NANSIG newsletter, emailed out to medical student subscribers across the UK. The survey link is also hosted on the Student Society of Myelopathy.org webpage and Facebook page.

### Eligibility and representation
All UK medical students are eligible to complete the survey. We aim to secure representation of all UK medical schools.

### Consent and confidentiality
A patient information sheet (PIS) including a question capturing informed consent forms the survey frontpage (figure 5). The PIS emphasises the rational, purpose, study aims and the voluntariness of participation. By necessity, the background information on DCM in the PIS is minimal to avoid compromising the DCM knowledge-based questions in the survey. A page block is used to separate the front and main survey pages.

Participants assign themselves a unique identifier based on a defined combination of characters from their mother's maiden name, the street they grew up on and their mobile phone number; this ensures anonymity and confidentiality while allowing the possibility of linking together future surveys from the same participant.

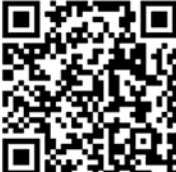

**Figure 4** A standard advertisement devised by the protocol development team and approved by the research ethics committee is being used for all survey advertising. Reproduced with permission from Myelopathy.org.

## Data security

All data are stored exclusively on the secure, online Qualtrics platform until closure of the survey. Thereafter, survey data will be extracted directly from the Qualtrics platform into a password-encrypted Excel spreadsheet (Microsoft, California, USA) on a password-protected computer. Only the immediate research team will be granted access to this on a need-to-access basis. After completion of data analysis, all data that are no longer required will be deleted. No participant-identifiable data will be collected or stored at any point.

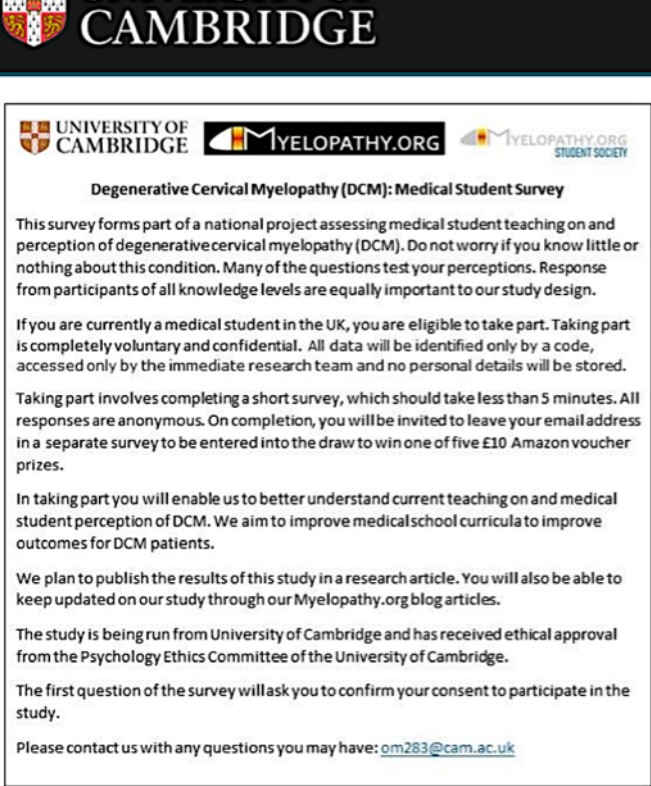

**Figure 5** Survey frontpage including patient information sheet and capturing of informed consent. Reproduced with permission from Myelopathy.org.

## Statistical analysis

Data will be analysed in Microsoft Excel and using statistical software.

Relationships between duration of DCM teaching, duration of private study on DCM and student perception of their DCM knowledge and objective performance in the DCM knowledge-based questions will be assessed. Differences in performance on the knowledge-based questions will be compared between groups based on medical school, year of medical school, duration of teaching, format of teaching, duration of private study, prior self-reported awareness of DCM, self-rating of DCM knowledge and between groups of medical students wishing to pursue neurological/neurosurgical career specialties and those who do not.

Due to all items in the survey being mandatory, we anticipate that missing data will be minimal, however will conduct a missing data analysis as necessary. It is not expected the data will be suitable for imputation, thus we will compare demographics to establish any selection bias caused by the exclusion of missing values. We will perform analysis on all collected values. The survey findings will be presented using descriptive statistics. Inferential statistical tests may be used to consider the relationship between knowledge and educational provision. Distribution testing will be performed. Appropriate statistical methods will be selected based on the class and distribution of the data; it is anticipated data will be non-parametric, and as measures are recorded on an ordinal or categorial basis, will therefore be compared using tests such as $\chi^2$ or Kruksall-Wallis tests.

## Patient and public involvement

Patients were involved in the conception, design, development and conduct of this study. A DCM Patient and Public Engagement Day hosted at the University of Cambridge was captured by Cambridge TV in a documentary.[11] During this day, a focus group of patients with DCM emphasised their experiences of diagnostic delays. This has been frequently echoed by patients in Myelopathy.org social media groups and in a DCM word-cloud creating initiative to understand patient's perceptions of the condition for a related project. Patients with DCM therefore inspired the conception of this study and approved the final survey design. Patients also assisted in the essential administrative tasks of the project, such as logo design. The online survey for the study was hosted on Myelopathy.org, which is a patient-maintained website for people with DCM. Patients were, therefore, active in aiding the dissemination of the survey to UK medical students. Patients who were involved in preparation of the manuscript are among the authors. Patients with DCM are involved in plans to disseminate this research to the patient community, including blog articles on Myelopathy.org, posts in online patient support groups and presence at spinal conferences in the UK.

## DISCUSSION

### Study rationale

This is the first study to describe the DCM education at medical schools nationally. Medical students are recruited to an online survey, including an assessment of their DCM knowledge and their personal DCM learning experience. By comparison of learning experience and knowledge, and also comparison between medical schools, it is anticipated both the presence or absence of a knowledge gap can be evaluated, and the more successful teaching formats are identified. This will be used to inform Myelopathy.org education recommendations.

### Why assess DCM education?

DCM is a devastating condition with one of the poorest quality of life scores of any chronic disease.[12] Symptoms include loss of manual dexterity, gait impairment, limb paraesthesia, limb weakness, limb stiffness, bladder and bowel disturbance, neck pain and neck stiffness.[1 13 14] The full spectrum of symptoms and the natural history of DCM are poorly appreciated,[15 16] yet it remains clear that the condition poses a substantial burden on the physical and mental health of people with DCM[17] and their supporters.[18] There is a broad differential for DCM symptoms[19] and poor consensus on what the most common DCM symptoms are. In addition, an increasing number of less well-described symptoms such as dysaesthetic sensory symptoms,[6] chest tightness[20] and headache[21] are being reported.[22] Neurological examination findings include upper motor neuron signs in the upper and lower limbs: spasticity and hyperreflexia in addition to weakness.[1 14]

The latest international guidelines[23] recommend surgical decompression as the only evidence-based management[24 25] for patients with moderate to severe, or progressive DCM. Surgery is also a management option in patients with mild DCM.[23] Importantly, surgery can halt progression of the condition but is ineffective at reversing existing damage.[25] Therefore, the traditional medical dogma that prevention is better than cure holds as strongly as ever in DCM: time is spine.[1]

Scientific advances are sorely needed to improve therapeutic options and ultimately to cure disabled patients with DCM.[26 27] However, while research into the neurobiology of DCM and neural regeneration,[28] such as the RECEDE-Myelopathy and CSM-PROTECT trials offer hope for the future,[9 29 30] even the most optimistic scientist would concede that tangible benefits for patients arising from these technologies remain on the horizon: one systematic review reported average duration of knowledge translation of 17 years from research to practice.[31]

Therefore, pragmatic approaches are required to improve outcomes for today's patients with DCM. One such approach entails addressing the substantial problem of diagnostic delays,[1] which is expected to improve outcomes in this progressive condition.[5] One method of instituting this may be through better education. Indeed, a phenomenon of 'neurophobia' is well described among medical students and non-specialist

doctors.[32–34] Additionally, evidence from our recent gap analysis suggests poor representation of DCM in medical curricula. These factors may point to a knowledge gap in DCM medical education.

## Benefits of a medical student survey

Assessing teaching and knowledge among medical students is a pragmatic and initial step to evaluate a knowledge gap. Medical school is the foundation of all medical professionals' education and is especially important for instilling basic knowledge of important areas of medicine that may fall outside a doctor's future field of specialisation. It is likely that this is particularly pertinent for knowledge that this not later core (ie, built on or refined) during specialisation. For this purpose, a survey of medical students is best placed to simultaneously assess knowledge, while reporting on the current provision of DCM teaching and perceptions of it. Moreover, the survey itself may help to prime an audience to support a subsequent intervention, or at the very least raise awareness.

Urgent improvements in outcomes for patients with DCM are needed. Earlier diagnosis would improve outcomes based on current treatments. This study aims to evaluate DCM education and knowledge, and explore their relationship, among medical students. It is hoped this will inform educational interventions that may offer the greatest chance of shaping the non-specialist doctor's diagnosis of DCM tomorrow.

## LIMITATIONS

Online surveys are an effective method of reaching students from diverse medical schools.[35] In addition, digital collection of data offers a secure and seamless stream, facilitating data analysis. However, their limitation is in understanding how representative the findings are.

In order to mitigate this, a number of different dissemination approaches are being employed. This includes the appointment of local representatives at each UK medical school, the use social media posts, society mailing lists, society newsletters, medical school email bulletins, medical school newsletters and also the use of prizes as an incentive.[36] Additionally, the capture of participant demographics will help to measure the sample characteristics.

The engagement with open surveys often involves some predefined interest. It is possible therefore a selection bias towards students interested in neuroscience is more likely and results would be considered as a best-case scenario. While there is a risk of overlooking or underestimating the magnitude of a knowledge gap given the outlined benefits of this methodology, this is an accepted risk. Whatever the relationship between DCM teaching and knowledge turns out to be, it will be instructive with regards to the overall objective of identifying ways of improving DCM diagnosis.

We acknowledge that the wording of the survey item 'Do you think time to diagnosis is important for long-term prognosis?' could be perceived as leading and would have been better phrased in a more neutral format such as 'How important is time to diagnosis for long-term prognosis?'. We will monitor the performance of this item closely in our analysis.

## ETHICS AND DISSEMINATION

Ethical approval for the study was granted by the Psychology Research Ethics Committee, University of Cambridge (PRE.2018.099). Insurance cover was arranged via the University of Cambridge (HVS/2018/2366). The findings of the study described in this protocol, and all other related work, will be submitted for publication in a peer-reviewed journal and will be presented at scientific conferences.

**Author affiliations**
[1]Division of Neurosurgery, Department of Clinical Neurosciences, University of Cambridge, Cambridge, UK
[2]School of Clinical Medicine, University of Cambridge, Cambridge, UK
[3]Ninewells Hospital and Medical School, University of Dundee, Dundee, UK
[4]Myelopathy.org, Cambridge, UK
[5]Neurology Unit, Department of Clinical Neurosciences, University of Cambridge, Cambridge, UK
[6]Anne McLaren Laboratory for Regenerative Medicine, Wellcome Trust-Medical Research Council Cambridge Stem Cell Institute, University of Cambridge, Cambridge, UK

**Acknowledgements** The authors are grateful to all the patients who have inspired this study, most especially at the University of Cambridge Patient and Public Engagement Day. In addition, the authors would like to thank the Student Society of Myelopathy.org and NANSIG medical student representatives for their assistance in disseminating the research survey.

**Contributors** OM, BD, IS, ES, SStacpoole and MK were involved in the conceptualisation and design of this study. OM wrote the first draft of the manuscript. OM, BD, MS, SSmith, AW, SA, MS, IS, ES, SStacpoole and MK reviewed and approved the final manuscript.

**Funding** Research in the MK's laboratory is supported by a core support grant from the Wellcome Trust and MRC to the Wellcome Trust–Medical Research Council Cambridge Stem Cell Institute. MK is supported by an NIHR Clinician Scientist Award and BD is supported by an NIHR Clinical Doctoral Research Fellowship.

**Disclaimer** This report is independent research arising from a Clinician Scientist Award, CS-2015-15-023, supported by the National Institute for Health Research. The views expressed in this publication are those of the authors and not necessarily those of the NHS, the National Institute for Health Research or the Department of Health and Social Care.

**Competing interests** OM, BD, MS, SSmith, AW, SA, MS, IS, ES and MK have voluntary roles at Myelopathy.org, an international DCM charity.

**Patient and public involvement** Patients and/or the public were involved in the design, or conduct, or reporting, or dissemination plans of this research. Refer to the Methods section for further details.

**Patient consent for publication** Not required.

**Provenance and peer review** Not commissioned; externally peer reviewed.

**ORCID iD**
Oliver Mowforth http://orcid.org/0000-0001-6788-745X

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
