## [Reviewer comments · BMJ Open]

ARTICLE DETAILS

TITLE (PROVISIONAL)	Current Provision of Myelopathy Education in Medical Schools in the United Kingdom: Protocol for a National Medical Student Survey
AUTHORS	Mowforth, Oliver; Davies, Benjamin; Stewart, Max; Smith, Sam; Willison, Alice; Ahmed, Shahzaib; Starkey, Michelle; Sadler, Iwan; Sarewitz, Ellen; Stacpoole, Sybil; Kotter, Mark

VERSION 1 - REVIEW

REVIEWER	Brian Heist University of Pittsburgh School of Medicine
REVIEW RETURNED	08-Dec-2019

GENERAL COMMENTS	This manuscript explains the protocol of a survey of medical students in the UK to help understand the current state of education addressing degenerative cervical myelopathy. The authors appropriately justify the survey and adequately explain its development and execution to enable repetition. Figure 3 contains a copy of the survey. I have two minor concerns as follows. 1. The dates of distribution of the survey should be listed, including any differences in timing among and within the targeted medical schools.2. There are some typos and minor oversights in the writing. These include: p. 5 Lines 22-24. I would combine the first two sentences. Line 39. Delete "of this" p. 10 Lines 15-17. Change relevant words to "dissemination of the survey" p. 11 Line 55. Change "improving" to "improve" Line 57. Change "may entail" to "entails" p. 12 Lines 17-19. I do not understand this sentence. Line 45. Remove comma. p. 13 Line 10. Change to "diagnosis of DCM tomorrow."
--

REVIEWER	Jeremy Moeller Department of Neurology Yale University
REVIEW RETURNED	17-Feb-2020

GENERAL COMMENTS	The authors describe the development and implementation of a national UK survey to determine students' perceptions of how much training they get about degenerative cervical myelopathy (DCM) and to relate this to knowledge about basic diagnostic and treatment considerations in this disorder. As the authors outline themselves, this protocol has several strengths, including the consistent involvement of content experts, students, other trainees and patients, and its national scope. The dissemination of the survey is discussed in adequate detail, and there were clearly attempts made to ensure the broadest possible reach. There are a few potential opportunities for improvement of this protocol, and (hopefully) of the eventual analysis and dissemination:  1. The authors should describe the development of their survey in greater detail.  - A useful guide to the steps of survey development can be found in Artino et al Med Teach 2014 (doi: 10.3109/0142159X.2014.889814). - For example, I would be interested in knowing more about why specific items were chosen. In the case of total duration of cervical myelopathy teaching, was the range from 0 min to 2 hours chosen because of the authors' estimate that the majority of respondents would have experiences that fall into this range? The process for choosing items for Part 2 was described in adequate detail. However, a few of the items may not have been optimally designed or worded. For example, a question like "Do you think time to diagnosis is likely important for long-term prognosis" is very susceptible to bias. The authors may have gotten more nuanced data (and data that would be more useful for their proposed subsequent teaching interventions) if they had chosen an alternative wording, like "How important is time to diagnosis for long-term prognosis" and then giving options on a 5-point scale from Not At All Important to Very Important. While it is not possible to change this item, it should probably be discussed as a potential limitation, and I would also advise reviewing this item carefully in the analysis phase, to determine if it has different performance characteristics than other items. - The pilot process should also be described in greater detail. It's fine to learn that the students found the survey acceptable and easy to use, but what about the pilot results? Did the data reveal any inconsistencies or ambiguous results? Was there any attempt to do cognitive interviews in order to ensure that the questions were interpreted by respondents in the way the authors intended? - Have there been similar national-scale medical student surveys regarding education and knowledge of other disease processes? The authors should consider adding a component of a literature search, and including some discussion of other such surveys, if they exist. What would be particularly relevant are surveys of similar types of diseases (e.g. those that are relatively common, under-recognized, and susceptible to harmful diagnostic delay). What are ways in which these surveys are similar or different than the current
--

	survey? - If some of the 7 steps outlined by Artino et al (2014) were not followed, they should be discussed as potential limitations of the study. 2. The authors should be more explicit about their proposed statistical analysis. I think it is reasonable to have a more specific plan about what statistical tools could be used to examine the relationship between self-reported DCM teaching and knowledge (e.g. descriptive statistics, ANOVA, factor analysis, etc.). By not being more explicit about the proposed statistical analyses they might use, the authors run the risk of choosing statistical tools for the purpose of ensuring that the data fit their pre-ordained conclusions. Based my reading of their introduction and discussion, it's clear that the authors suspect that there is very little teaching about DCM in UK medical schools, and they suspect knowledge will be low overall. Regardless of this suspicion, they should go into the analysis open to the possibility of unexpected results. The authors could also comment on what types of analyses they were able to do with the 20-person pilot data: not so much for the purposes of deriving any conclusions, but rather to get a sense of what types of tests are feasible and appropriate. 3. The exact time range of survey administration should be stated explicitly (start date to end date).
--	--

VERSION 1 – AUTHOR RESPONSE

Reviewer(s)' Comments to Author:

Reviewer: 1

Reviewer Name: Brian Heist

Institution and Country: University of Pittsburgh School of Medicine

Please state any competing interests or state 'None declared': None declared

Please leave your comments for the authors below

This manuscript explains the protocol of a survey of medical students in the UK to help understand the current state of education addressing degenerative cervical myelopathy. The authors appropriately justify the survey and adequately explain its development and execution to enable repetition. Figure 3 contains a copy of the survey.

A: We thank Dr Heist for kindly reviewing our manuscript

I have two minor concerns as follows.

1. The dates of distribution of the survey should be listed, including any differences in timing among and within the targeted medical schools.

A: We have added additional detail on the timing of survey advertisements and have clarified that this schedule was the same for all medical schools.

2. There are some typos and minor oversights in the writing. These include:

p. 5 Lines 22-24. I would combine the first two sentences.

Line 39. Delete "of this"

p. 10 Lines 15-17. Change relevant words to “dissemination of the survey”

p. 11 Line 55. Change “improving” to “improve”
Line 57. Change “may entail” to “entails”

p. 12 Lines 17-19. I do not understand this sentence.
Line 45. Remove comma.

p. 13 Line 10. Change to “diagnosis of DCM tomorrow.”

A: We thank Dr Heist for kindly highlighting these errors, which we have corrected.

We are grateful to Dr Heist for reviewing our manuscript and for the improvement as a result of making the suggested changes.

Reviewer: 2

Reviewer Name: Jeremy Moeller

Institution and Country: Department of Neurology

Yale University

Please state any competing interests or state ‘None declared’: None declared

Please leave your comments for the authors below

The authors describe the development and implementation of a national UK survey to determine students' perceptions of how much training they get about degenerative cervical myelopathy (DCM) and to relate this to knowledge about basic diagnostic and treatment considerations in this disorder.

As the authors outline themselves, this protocol has several strengths, including the consistent involvement of content experts, students, other trainees and patients, and its national scope. The dissemination of the survey is discussed in adequate detail, and there were clearly attempts made to ensure the broadest possible reach.

There are a few potential opportunities for improvement of this protocol, and (hopefully) of the eventual analysis and dissemination:

A: We thank Dr Moeller for kindly reviewing our manuscript and for highlighting the strengths of the protocol.

1. The authors should describe the development of their survey in greater detail.

- A useful guide to the steps of survey development can be found in Artino et al Med Teach 2014 (doi: 10.3109/0142159X.2014.889814).

A: We thank Dr Moeller for sharing this highly informative reference, which we read with great interest. We believe that there is a lot of work to be done in the field of myelopathy medical education, hence we anticipate this guide being valuable in the design of follow-up surveys. We are supportive of the ideology of standardisation in this guide, which we are currently promoting more widely in DCM research through the RECODE-DCM initiative (<https://recode-dcm.com/>)

We feel that all steps have been adequately addressed in our study, with the exception of a systematic review on current research on DCM medical education, which is not possible due to the current lack of published research in this area. However, we feel we addressed this with the next best possible method in the form of our recently published quantitative analysis of UK medical student

knowledge of DCM (DOI: 10.1136/bmjopen-2018-028455). We do agree that a systematic review of medical student surveys in other diseases would be valuable to learn lessons to apply to our future work.

- For example, I would be interested in knowing more about why specific items were chosen. In the case of total duration of cervical myelopathy teaching, was the range from 0 min to 2 hours chosen because of the authors' estimate that the majority of respondents would have experiences that fall into this range? The process for choosing items for Part 2 was described in adequate detail. However, a few of the items may not have been optimally designed or worded. For example, a question like "Do you think time to diagnosis is likely important for long-term prognosis" is very susceptible to bias. The authors may have gotten more nuanced data (and data that would be more useful for their proposed subsequent teaching interventions) if they had chosen an alternative wording, like "How important is time to diagnosis for long-term prognosis" and then giving options on a 5-point scale from Not At All Important to Very Important. While it is not possible to change this item, it should probably be discussed as a potential limitation, and I would also advise reviewing this item carefully in the analysis phase, to determine if it has different performance characteristics than other items.

A: Our survey items were chosen following consultation and discussion between the following group of individuals, who each brought a unique area of expertise: an academic consultant neurosurgeon specialising in DCM, an academic consultant neurologist specialising in medical education, an academic neurosurgical trainee with expertise of both DCM and postgraduate training in the UK, medical students with expertise on current teaching provisions at multiple medical schools in the UK, representatives from the international charity Myelopathy.org with expertise on myelopathy and patients with lived experience of DCM. Items were designed by the above team to (1) capture student demographic details and (2) to capture medical student knowledge of the basic facts that we collectively agreed were a set of essential DCM facts that all doctors should appreciate. A pragmatic, experience-based approach, was taken to select the response options for the survey items. For example, based on many years of supervising and teaching students from medical schools across the UK, we estimate that most UK medical students' DCM teaching will fall between 0-2 hours. We do acknowledge the risk of bias and will monitor for this closely in the data. We deliberately left the end response open to accommodate all values above our range and we will look closely at the data to assess the frequency of this response. This will inform future work.

We acknowledge the risk of bias as a result of the wording of the above survey item, have acknowledged this as a limitation within this protocol and will monitor the performance of this item carefully.

- The pilot process should also be described in greater detail. It's fine to learn that the students found the survey acceptable and easy to use, but what about the pilot results? Did the data reveal any inconsistencies or ambiguous results? Was there any attempt to do cognitive interviews in order to ensure that the questions were interpreted by respondents in the way the authors intended?

A: No formal statistical analysis was conducted on our pilot results. Data were inspected closely for unexpected, ambiguous or inconsistent results. The survey data was consistent, and the survey had been completed as intended. All students in the pilot group were contacted to ask if they had encountered any difficulties, areas of uncertainty or had suggestions for improving the survey. No areas of difficulty or suggestions for improvement were reported. On this basis we proceeded to the main phase of the survey. We have expanded on this in the manuscript.

- Have there been similar national-scale medical student surveys regarding education and knowledge of other disease processes? The authors should consider adding a component of a literature search,

and including some discussion of other such surveys, if they exist. What would be particularly relevant are surveys of similar types of diseases (e.g. those that are relatively common, under-recognized, and susceptible to harmful diagnostic delay). What are ways in which these surveys are similar or different than the current survey?

A: We had not considered this but agree it could offer some interesting perspectives. We are not aware of any such surveys in the UK but agree that this could be more comprehensively considered with a literature review and would be helpful to inform strategies going forward. At this stage, we feel such a programme of work is beyond the scope of this specific protocol paper. However, we have instead elected to establish this work in parallel – we would be delighted and open to collaborate on this with Dr Moeller, if so interested.

- If some of the 7 steps outlined by Artino et al (2014) were not followed, they should be discussed as potential limitations of the study.

• A: We were not familiar with the 7 steps outlined by Artino et al (2014), however our methodology closely aligns with it. Only Step 1 was not specifically addressed as recommended. Rather than a systematic review, the foundations of this study lie in a quantitative gap analysis of medical students' and physicians' knowledge of DCM, in which we compared DCM to other conditions in terms of disease-specific content in common medical education materials and student performance on disease-specific questions in online revision banks. This was recently published (DOI: 10.1136/bmjopen-2018-028455) and included a literature review. Although perhaps not as focused as a systematic review, we do feel this foundation will have served a similar purpose, particularly given the other 6 steps were addressed adequately by our methodology. We have added a concise discussion of the above to the manuscript.

2. The authors should be more explicit about their proposed statistical analysis. I think it is reasonable to have a more specific plan about what statistical tools could be used to examine the relationship between self-reported DCM teaching and knowledge (e.g. descriptive statistics, ANOVA, factor analysis, etc.). By not being more explicit about the proposed statistical analyses they might use, the authors run the risk of choosing statistical tools for the purpose of ensuring that the data fit their pre-ordained conclusions. Based my reading of their introduction and discussion, it's clear that the authors suspect that there is very little teaching about DCM in UK medical schools, and they suspect knowledge will be low overall. Regardless of this suspicion, they should go into the analysis open to the possibility of unexpected results. The authors could also comment on what types of analyses they were able to do with the 20-person pilot data: not so much for the purposes of deriving any conclusions, but rather to get a sense of what types of tests are feasible and appropriate.

A: The pilot phase was used to check the survey was performing as intended without error or areas of ambiguity. No formal statistical analysis was conducted. We have clarified this within the methods section. We approach the analysis with an open mind and will report the results honestly whatever they show – we very much hope to find that UK myelopathy medical education is better than we hypothesise! Despite its underrepresentation in medical and post graduate curricula and training resources, this was our experience when analysing the performance of students in an online question bank (DOI: 10.1136/bmjopen-2018-028455)

Due to all items in the survey being mandatory, we anticipate that missing data will be minimal, however will conduct a missing data analysis as necessary. It is not expected the data will be suitable for imputation, thus we will compare demographics to establish any selection bias caused by the exclusion of missing values. We will perform analysis on all collected values. The survey findings will be presented using descriptive statistics. Inferential statistical tests may be used to consider the relationship between knowledge and educational provision. Distribution testing will be performed; it is

anticipated data will be non-parametric, and as measures are recorded on an ordinal or categorical basis, will therefore be compared using tests such as Chi-Squared or Kruksall Wallis Tests.

We have added a concise discussion of the above to the manuscript.

3. The exact time range of survey administration should be stated explicitly (start date to end date).

A: We have added this detail accordingly.

We thank Dr Moeller for comprehensively reviewing our manuscript and are grateful for the improvements that have emanated from this review. We are particularly grateful to Dr Moeller for sharing the excellent Artino et al., 2014 reference, which will be highly valuable to our future work.

VERSION 2 – REVIEW

REVIEWER	Jeremy Moeller Yale University School of Medicine USA
REVIEW RETURNED	05-Apr-2020

GENERAL COMMENTS	I am satisfied with the changes made to address prior concerns regarding this publication. I will be interested to read of their results when they become available.
--